# Generation of Transgenic Sperm Expressing GFP by Lentivirus Transduction of Spermatogonial Stem Cells In Vivo in Cynomolgus Monkeys

**DOI:** 10.3390/vetsci10020104

**Published:** 2023-02-01

**Authors:** Shengnan Wang, Yanchao Duan, Bingbing Chen, Shuai Qiu, Tianzhuang Huang, Wei Si

**Affiliations:** 1State Key Laboratory of Primate Biomedical Research, Institute of Primate Translational Medicine, Kunming University of Science and Technology, Kunming 650500, China; 2Yunnan Key Laboratory of Primate Biomedical Research, Kunming 650500, China

**Keywords:** cynomolgus macaque, spermatogonial stem cells, enhanced green fluorescent protein, transgenic sperm

## Abstract

**Simple Summary:**

Gene editing technology is expected to be an effective way to establish animal models of human diseases. Using spermatogonial stem cells and sperm as gene editing objects is simple to operate. However, there has been no research on gene editing of spermatogonial stem cells to obtain genetically modified sperm, and it is not known whether cryopreservation has adverse effects on genetically modified sperm. Furthermore, there are few clear cell surface markers to identify primate spermatogonial stem cells. In this experiment, we used ultrasound to guide the injection needle into the testis reticulum space to inject the enhanced green fluorescent protein lentivirus from the testis reticulum into seminiferous tubules. Finally, we successfully obtained transgenic sperm, which have a similar freezing and recovery rate to that of wild animals. This approach is expected to be another effective way to establish edited animal disease models, and will play an important role in the research of human diseases and the development of new drugs and therapeutic methods.

**Abstract:**

Nonhuman primates (NHPs) have been considered as the best models for biomedical research due to their high similarities in genomic, metabolomic, physiological and pathological features to humans. However, generation of genetically modified NHPs through traditional methods, such as microinjection into the pronuclei of one-cell embryos, is prohibitive due to the targeting efficiency and the number of NHPs needed as oocyte/zygote donors. Using spermatogonial stem cells (SSCs) as the target of gene editing, producing gene-edited sperm for fertilization, is proven to be an effective way to establish gene editing animal disease models. In this experiment, we used ultrasound to guide the echo dense injection needle into the rete testis space, allowing the EGFP lentivirus to be slowly injected at positive pressure from the rete testis into seminiferous tubules. We found Thy1 can be used as a surface marker of cynomolgus monkey SSCs, confirming that SSCs carry the GFP gene. Finally, we successfully obtained transgenic sperm, with a similar freezing and recovery rate to that of WT animals.

## 1. Introduction

Experimental animals have promoted the development and progress of biomedical research. However, the lack of appropriate animal models that can highly simulate human pathogenesis and pathology is the bottleneck of disease mechanism study and drug development [1]. The most widely used rodent models cannot ideally replicate human pathological features. For example, age-associated glucose homeostasis trends differ between mice and monkeys/humans [2]. In contrast, non-human primates (NHPs) exhibit similarities with humans in genetic material, physiological characteristics, immune system and other aspects. NHPs may be the most ideal animal model for human disease research, drug development, and therapeutic strategy verification [3,4]. Animals with spontaneous genetic mutations present greatly similar pathogenesis and phenotypes to humans, but the probability of spontaneous genetic mutations in NHP colonies is extremely low. Although specific NHP disease models can be quickly generated by chemical induction in most cases, these models cannot accurately reflect the pathogenesis of human diseases.

The rapid development of gene editing technology enables the efficient creation of animal models with either spontaneous mutations or targeted mutations. Lentivirus is a very effective gene delivery vector and is widely used in the generation of mice, rats, rabbits and transgenic NHP models [5,6,7,8]. For example, transgenic NHP models of autism and Huntington’s disease have been generated by over-expressing *MeCP2* and *HTT* genes via lentiviral transduction of zygotes, respectively [9,10,11]. In addition, autism NHP models have been generated by *SHANK3* gene knockout via CRISPR-Cas9 [12]. In general, these reported gene editing for the generation of NHP models either by using lentivirus or CRISPR-Cas9 and all needed microinjection on monkey oocytes or zygotes [13,14,15,16], which need a lot of monkeys as oocyte donors and embryo transfer recipients due to low efficiency.

In contrast, male monkeys produce a great number of sperm. Therefore, genetic modification of male SSCs can be an alternative transgenic strategy to generate gene editing monkey models without delicate micromanipulation or high demand for female monkey resources [17]. SSCs maintain spermatogenesis throughout a male’s lifespan via self-renewal and differentiation in vivo [18]. Once gene editing can be achieved in SSCs, the genetic modification in the germline allows the stable delivery of the mutant characteristics to the descendants [19]. Although the first generation of genetically modified animals via genetically modified SSCs is heterozygous, homozygous offspring can be obtained via breeding. Meanwhile, sperm are easy for cryopreservation, which benefits genetic resource preservation, genetic resource sharing and mass production of offspring [20]. Transgenic sperm have been produced in vivo by lentivirus transduction of SSCs and resulted in subsequently transgenic progeny in rodents [21]. In contrast to rodents, the isolation, purification and identification of SSCs remain unsuccessful in primates (monkey and human), which is mainly due to that there are barely any clear cell surface markers to identify primate SSCs [22]. Furthermore, the culture conditions for primate SSCs are not optimal, and it is impossible to culture and expand primate SSCs in vitro [23,24,25]. Consequently, it is difficult to perform gene manipulation on primate SSCs in vitro. Therefore, the application of lentivirus as a vehicle to transfer exogenous genes into SSC genomes in vivo and generate transgenic sperm is expected to become an alternative and practicable strategy to generate disease models in NHPs.

The cost for generation and maintenance of transgenic and gene-edited monkeys is expensive. Semen cryopreservation provides an efficient way to preserve valuable genetic resources, and offspring can be easily produced by the application of assistant reproductive technology such as artificial insemination. The cryopreservation process is known to cause several types of damage to sperm survival and integrity. Previous study indicated that expression of exogenous gene in mouse testis resulted in decreased sperm motility and fertility [26,27,28], which may depend on the transgene and expression efficiency. Therefore, we also examined sperm motility, acrosomal integrity and mitochondria membrane potential in sperm collected before and 2 months after EGFP lentiviral transfection [20].

The goal of this study was to create genetically modified primate SSCs via lentiviral transduction. Lentiviruses carrying enhanced green fluorescence protein (EGFP) were injected into the rete testis of cynomolgus monkeys. The integration of exogenous EGFP genes was examined by the expression of EGFP in SSCs and qPCR. Finally, we successfully obtained transgenic sperm which have similarly freezing resuscitation rate to that of WT animals. The completion of this pilot study will provide a proof of principle for using SSCs as an alternative for generating NHP models via genetic modifications of their SSCs.

## 2. Materials and Methods

### 2.1. Animals and Ethics

Three male cynomolgus macaques (6–10 years old) with testicular lentivirus vector injection and sperm collection and one male cynomolgus macaque (9 years old) for trypan blue injection were provided by the State Key Laboratory of Primate Biomedical Research (Kunming, Yunnan, China). All procedures were approved by the Institutional Animal Care and Use Committee of Kunming University of Science and Technology (Authorization code: LPBR202104015), and were carried out in accordance with the 8th edition of the “Guide for the Care and Use of Laboratory Animals”.

### 2.2. EGFP Lentivirus

Lentiviral vectors carrying EGFP were purchased from Shanghai Genechem Co., Ltd. (Shanghai, China, http://www.genechem.com.cn (accessed on 5 June 2021)). The EGFP lentivirus were stored in a −80 °C freezer and thawed on ice before use. The carrier information was: hU6-MCS-Ubiqutin-EGFP-IRES-puromycin. Fifty µL of EGFP lentiviruses (1.0 × 10^9^ TU/mL) was diluted to 1 mL with saline and then injected into the rete testicle of cynomolgus monkeys.

### 2.3. Injection of EGFP Lentivirus into the Rete of Cynomolgus Macaque Testicles

Cynomolgus macaques were anesthetized with a 10 mg/kg ketamine chloride intramuscular injection. The rete showed a linear echo dense structure under ultrasound (GE Healthcare, LOGIQ E10). The EGFP lentivirus diluent was loaded into a syringe with a needle (KINDLY GROUP, Shanghai, 0.7 × 80 TWLB, www.kdlchina.cn, accessed on 5 June 2021.)) and then injected into the rete of both testicles under the guidance of an ultrasound system. The needle bevel was punctured into the rete of testicular tissue and EGFP lentivirus vehicles were injected from the epididymis head into the rete [29].

### 2.4. Pathology of Testicular Tissue

In order to confirm whether the injected EGFP lentivirus integrated into SSCs during spermatogenesis in cynomolgus monkeys, we compared the morphology of testicular tissues before and 2 months post EGFP lentivirus injection since the duration of spermatogenesis from SSCs to sperm in cynomolgus macaque is roughly 42–44 days, followed by 10.5 days of epididymal transport time [30,31]. The seminiferous tubules tissues obtained by biopsy were directly fixed with 4% Paraformaldehyde (PFA) for 2 days, embedded into paraffin (Thermo Fisher Scientific, Waltham, MA, USA) and sliced into 5 μm sections. The samples were then embedded into paraffin (Thermo Fisher Scientific, Waltham, MA, USA) and sliced into 5 μm sections. The sections were deparaffinized in xylene twice for 10 min each time, and then were put in 100%, 100%, 95%, 80%, 75%, 60%, and 50% ethanol for 10 min each. Then, the sections were stained with hematoxylin for 3 min and washed with PBS 3 times (5 min each). Subsequently, the sections were soaked in ethanol (1% HCL) for 10 s and rinsed with PBS for 3 times, and in eosin for 1 min and soaked in 50%, 60%, 75%, 80%, 95%, and 100% ethanol for 10 min. Finally, the sections were rinsed with anhydrous ethanol and soaked in xylene for 30 min before being sealed and examined with a light microscope. We selected 3 sections from each monkey of three male cynomolgus macaques and analyzed 3 seminiferous tubules in each section.

### 2.5. Immunofluorescence Examination

For immunofluorescence examination, the embedded samples were sliced into 10 μm sections., testicular sections were washed for 10 min with Phosphate Buffer Saline (PBS) containing 0.3% trition and blocked with 3% Bovine Serum Albumin (BSA) at room temperature for 1 h. To differentiate SSCs from other types of cells, Thy1 was selected as the marker of NHP SSCs, as Thy1 was successfully used as a marker for sorting SSCs from tree shrews, a close relative to primates, and transgenic tree shrews were created through genetic modification of the sorted SSCs [19]. Slides were incubated with primary antibody directed against Thy1(1:50) (Life Span Bio Sciences, Washington, CA, USA) at 4 °C overnight; the samples were washed three times with PBS, followed by incubation with secondary antibodies(1:500) (Thermo Fisher Scientific, Waltham, MA, USA) for 1 h at room temperature and were washed with PBS. The slides were then examined using a Leica TCS SP8 DIVE confocal microscope. Images for Thy1 indirect immunofluorescence were captured at excitation wavelength: 593 nm and emission wavelength: 618 nm. Images for EGFP fluorescence were captured at excitation wavelength: 488 nm; emission wavelength: 509 nm.

### 2.6. Semen Collection, Evaluation and Cryopreservation

Eight semen samples collected from each of the three male monkeys were performed for cryopreservation and evaluation. The TRIS-egg yolk-based sperm freezing medium (TTE) containing 0.2% Tris, 1.2% TES, 2% glucose, 2% lactose, 0.2% raffinose, and 20% (*v/v*) fresh egg yolk was prepared as previously described [32,33]. Before an experiment, the medium was thawed in a 37 ℃ water bath. Semen was collected from the male monkeys by penile electroejaculation [34]. The volume of the remaining semen was measured, and TTE solution of equal volume containing 20% glycerol was prepared and placed at 4 °C. After 2 h, a TTE solution containing 20% glycerol was slowly added into the semen sample [33]. The diluted semen was sealed in 0.25 mL cryostraws and at 4 cm above liquid nitrogen with vapor for 10 min before being placed in liquid nitrogen and was stored at −196 °C until use. The cryopreserved semen were thawed in a 37 °C water bath.

Sperm motility, sperm acrosome integrity and mitochondrial membrane potential of fresh and frozen–thawed sperm were evaluated as in our previous description [20]. Fresh sperm and thawed sperm samples (10 μL) were placed on a pre-warmed Makler counting chamber (Sefi Medical Instruments, Haifa, Israel) under a microscope for motility assessment. At least 200 sperm of each sample were evaluated for motility. The sperm acrosome integrity for fresh and thawed samples was determined using the Alexa Fluor-488-peanut agglutinin conjugate assay (Eugene, OR, USA). Briefly, 10 μL fresh semen or frozen–thawed semen was evenly coated on a slide, and then was dried at room temperature and fixed for 30 min (200 μL anhydrous ethanol). Then 50 μL of 10 μg/mL Alexa Flu-488-peanut lectin was added, and the slides were incubated in a 37 °C dark box. Thirty minutes later, they were observed using a microscope (emission ratio of 530 nm, wavelength of 488 nm). Sperm with uniform green fluorescence in the acrosome region of the head were considered to have an intact acrosome, while sperm with little or no green fluorescent staining in the front of the head were considered to have acrosomal sperm impairment. At least 200 sperm per semen sample were evaluated for this staining. We used a JC-1 assay kit (Solarbio, Beijing, China) to detect mitochondrial membrane potential in 2 × 10^5^ fresh sperm and thawed resuscitated sperm. JC-1 reagent was incubated with semen in a 37 °C water bath for 20 min according to the instructions, and fluorescence detection was performed (emission ratio of 530 nm, wavelength of 488 nm). Sperm with intact mitochondria fluoresce in orange and yellow. In contrast, sperm with damaged mitochondria emit green fluorescence. At least 200 sperm in each sample were evaluated for mitochondrial potential using a fluorescent staining procedure.

### 2.7. Sperm DNA Isolation, q-PCR and Sequencing

To assess the amount of EGFP gene integration in the genome, a sperm sample containing 5 × 10^6^ sperm was centrifuged at 9500× *g* for 1 min, and then the supernatant was discarded. DNA was extracted using TIANamp Genomic DNA kit from TIANGEN (Beijing, China) according to the manufacturer’s instructions. PCR amplification of the integrated EGFP gene was conducted by using a primer pair (5′-GGCAAGCTGACCCTGAAGTT-3′; 5′-TCTCGTTGGGGTCTTTGCTC-3′). The PCR products were sequenced by using the first-generation sequencing technique according to Tsingke Biotechnology Co., Ltd. (Qingdao, China). (ABI 3730 xl). The qPCR was performed with SYBR Green PCR master mix (Roche, Switzerland). Relative expression of DNA was normalized to β-actin and calculated by 2^−ΔΔCt^.

### 2.8. Statistical Analysis

All data are expressed as mean ± standard deviation. The results of the homogeneity test of variances show that the internal variances of each group are equal. Sperm motility, acrosome integrity, and percentage of mitochondrial membrane potential were measured by variance analysis and the Fisher protection minimum significance difference test (SPSS 16, SPSS, Chicago, IL, USA), with a *p* value less than 0.05 considered statistically significant.

### 2.9. Data Availability

Data are to be shared upon request to the corresponding author.

## 3. Results

### 3.1. Validation of Rete Testis Injection Method and Introduction of EGFP Lentiviruses

The experimental timeline was shown in Figure 1A. To demonstrate the validity of the injection, firstly, we injected one monkey with 1 mL trypan blue staining solution (0.4%) into rete testis, instead of lentivirus. The injected testis was surgically removed and successful injection was confirmed by the dilatation of the rete testis space and scattering of fluid from the tubules to the surrounding seminiferous tubules and the ductules of the caput epididymis. Testicular tissue was cut in half after injection of Trypan blue, which accounts for about 60–80% of the seminiferous tubules (Figure 1B,C). After validation with 0.4% trypan blue solution, lentiviruses carrying EGFP were introduced into the rete testis and seminiferous tubules of three cynomolgus macaques using the same approach (Appendix A).

### 3.2. The Injection of EGFP Lentivirus Did Not Affect Spermatogenesis

The HE staining showed no obvious change in either SSCs located at the basement membrane of seminiferous tubules or sperm cells in the middle of the tubules. So EGFP lentivirus injection did not affect the morphology of testicular tissue and spermatogenesis (Figure 2A,B), and the original image is shown in Appendix A.

### 3.3. EGFP Expressed in SSCs

The green fluorescence signals of EGFP were observed in the biopsied seminiferous tubule tissue under a confocal microscope (Figure 3A). The SSCs were identified by the specific surface marker Thy1, and the results showed that NHP SSCs successfully carried the EGFP gene and expressed EGFP (Figure 3B). We found specific immunolabeling was observed around the wall of the seminiferous tubules. In order to confirm that Thy1 could be the marker of SSCs and not the result of secondary antibodies, Thy1-secondary antibodies were stained for the EGFP-transfected seminiferous tubules and no fluorescence was found. Therefore, Thy1 could be used as the marker of SSCs (Appendix A).

### 3.4. Confirmation of the Generation of Transgenic Sperm and Sperm Cryopreservation

Two months after EGFP lentivirus injection, we collected sperm once a week for eight weeks. PCR was used to detect the integration of the EGFP gene in the sperm genome at eight weeks after EGFP lentivirus injection. The sequence alignment of the EGFP gene was performed using a first-generation sequencing technique at the 8th week, and the alignment rate was 99%, which demonstrated that the EGFP gene was integrated into the sperm genome (Figure 4A, Appendix A and Figure 5). The amount of EGFP integration in the genome was measured by q-PCR; the expression level of the EGFP gene was 8.69 pg/mL in Monkey1, 1.55 pg/mL in Monkey 2 and 3.35 pg/mL in Monkey 3 (Figure 4B and Appendix A). No differences in sperm motility (Figure 4C and Appendix A), acrosomal integrity and mitochondria membrane potential (Figure 4D–G, Appendix A) were observed between sperm collected before and 8 weeks after EGFP lentivirus injection (*p* > 0.05).

## 4. Discussion

In the present study, we investigated the feasibility to create genetically modified primate SSCs via lentiviral transduction. The preliminary results demonstrated that genetically modified SSCs can be generated through injection of lentiviral vectors into the rete testicle of NHPs. The integration of exogenous genes did not significantly compromise the process of spermatogenesis, and sperm (including EGFP-positive) with normal morphology and motility were obtained.

SSCs are the only stem cells in postnatal male animals that can transmit genetic information to the offspring. The transduction of SSCs provides a new approach for animal transgenesis. The modification of SSCs by lentiviral transduction in vitro in rodents can achieve about 50% transgenic offspring [35,36]. However, the efficiencies were very low in other species using the same approach. For example, the transgenic tree shrew progeny resulting from lentiviral transduction in SSCs were only 5.7% [19]. Nevertheless, a similar approach in NHPs has never been reported due mainly to the unsatisfactory in vitro culture system for SSCs. In the present study, we injected EGFP lentiviruses into testicular reticulum of cynomolgus macaques and successfully obtained transgenic sperm. The integration of the exogenous EGFP gene into the sperm genome of cynomolgus macaques has been detected and confirmed by q-PCR and DNA sequencing. However, we used 20 donated cynomolgus oocytes for the fertilization assay, which was performed as in the previous description [3], but none expressed EGFP in the subsequent embryo cleavage and development (data not shown). The result indicated that the ratio of sperm carrying the EGFP gene was low.

The cryopreservation process is known to cause several types of damage to sperm survival and integrity. In our previous study, adding a cryopreservation agent to normal cynomolgus monkey semen reduced sperm cell damage caused by freezing and improved fertilization ability [20]. With the development of transgenic technology in male animals, research on cryopreservation of transgenic sperm may provide new insights into the fertility of transgenic animals. Previously, Huntington’s disease semen have similar rates to that of WT on the basis of motility, membrane integrity, and acrosome integrity [37]. Our results were similar to the Huntington’s model sperm in motility, acrosome integrity and mitochondrial membrane potential, further indicating that the TRIS-egg yolk-based sperm freezing medium is effective for the cryopreservation of transgenic sperm. At present, there are few reports on cryopreservation of transgenic animals, but further optimization of cryopreservation of transgenic sperm is necessary to improve the survival rate after thawing, which is crucial for the establishment of sperm banks. Although our preliminary experiment failed to obtain transgenic offspring, it showed that transgenic sperm viability, acrosome integrity and mitochondrial membrane potential were not affected. This preserves species resources for expanding transgenic animal populations. The transgenic sperm will produce genetically modified NHP in the future. The preliminary data from the experiments showed that EGFP transgenes were successfully integrated into the primate sperm genome after injection of viral vectors into the rete testicle. These experiments demonstrated that transgenesis in NHPs can be achieved through in vivo lentiviral transduction of SSCs. These results have important implications for using SSC targeting as an alternative and practicable strategy for nonhuman primate model creation. Therefore, this approach is expected to become another effective way to establish gene edited animal disease models, and play an important role in the research of male reproductive diseases and the development of new drugs and therapeutic methods.

## 5. Conclusions

We successfully obtained transgenic sperm by transfecting in vivo spermatogonial stem cells with lentivirus, and the motility, acrosome integrity and mitochondrial membrane potential of the transgenic sperm were similar to those of the wild type, indicating that the intervention of foreign genes did not affect the sperm structure.

## Figures and Tables

**Figure 1 vetsci-10-00104-f001:**
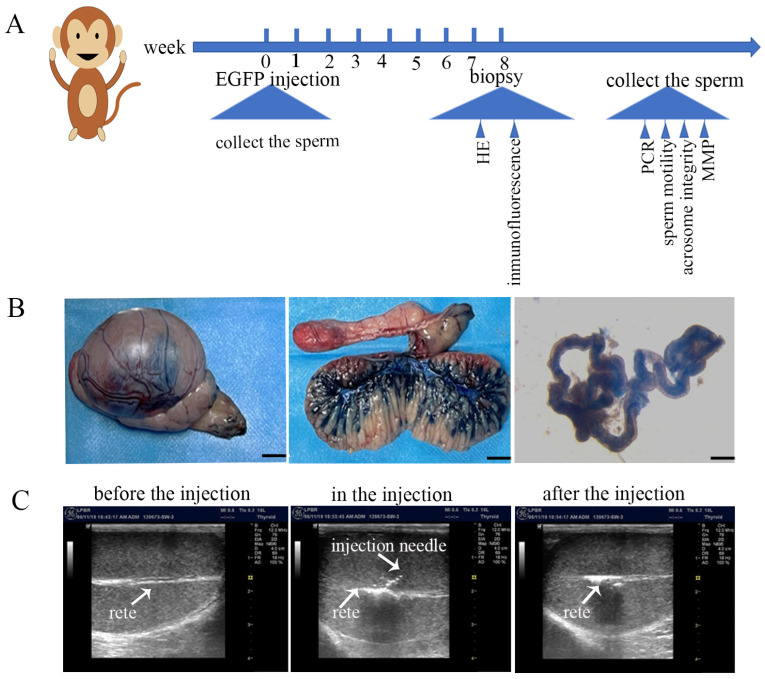
EGFP lentivirus injection: (**A**) Experimental timeline for EGFP lentivirus injection, testicular tissue and sperm collection; (**B**) Injecting of trypan blue staining solution (0.4%) or EGFP lentivirus into the rete testis and seminiferous tubules. The dilatation of rete after injection was confirmed by ultrasound. The rete and injection needle are indicated by the arrows; (**C**) Confirmation of successful injection of Trypan blue into rete testis. Bisection of the injected testis revealed that blue dye radiated rom the rete testis into approximately 60–80% of seminiferous tubules (*n* = 3).

**Figure 2 vetsci-10-00104-f002:**
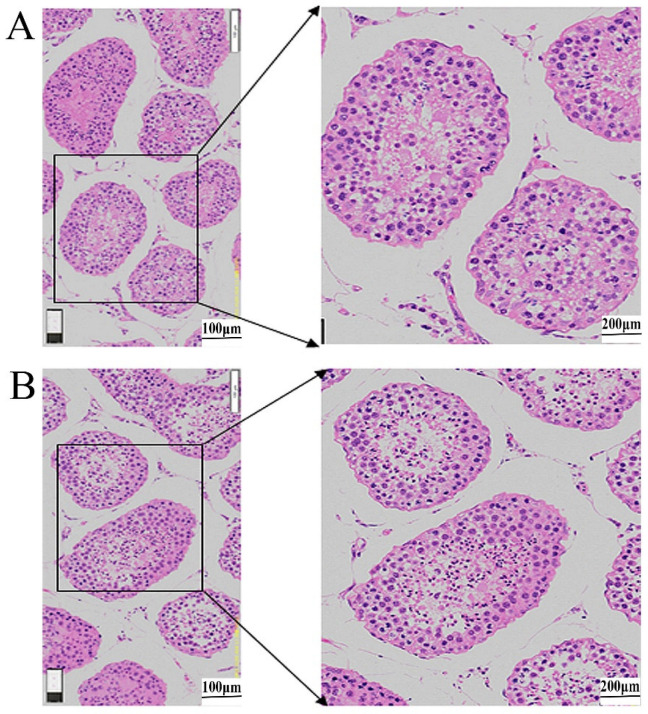
EGFP lentivirus does not affect the morphology of testicular tissue and spermatogenesis. (**A**) Testis pre-EGFP: Testicular tissue morphology at different microscope magnifications before EGF lentivirus injection. (**B**) Testis after EGFP: Testicular tissue morphology at different microscope magnifications after EGFP lentivirus injection (*n* = 3).

**Figure 3 vetsci-10-00104-f003:**
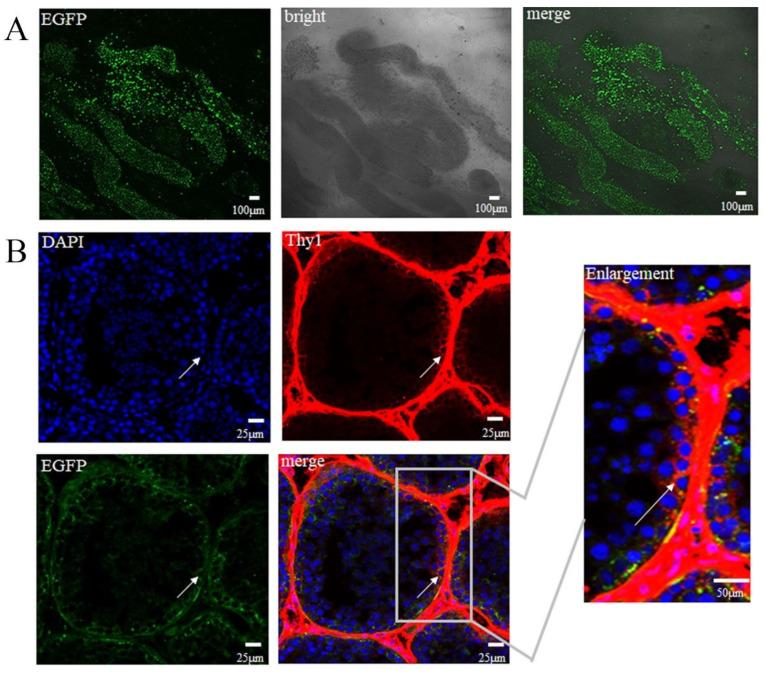
SSCs successfully carried the *EGFP* gene. (**A**) Green fluorescence signal of EGFP was observed in the seminiferous tubule tissue. (**B**) Thy1, a specific marker on the surface of SSCs; SSCs carried the EGFP gene (*n* = 3). The arrows represent SSCs.

**Figure 4 vetsci-10-00104-f004:**
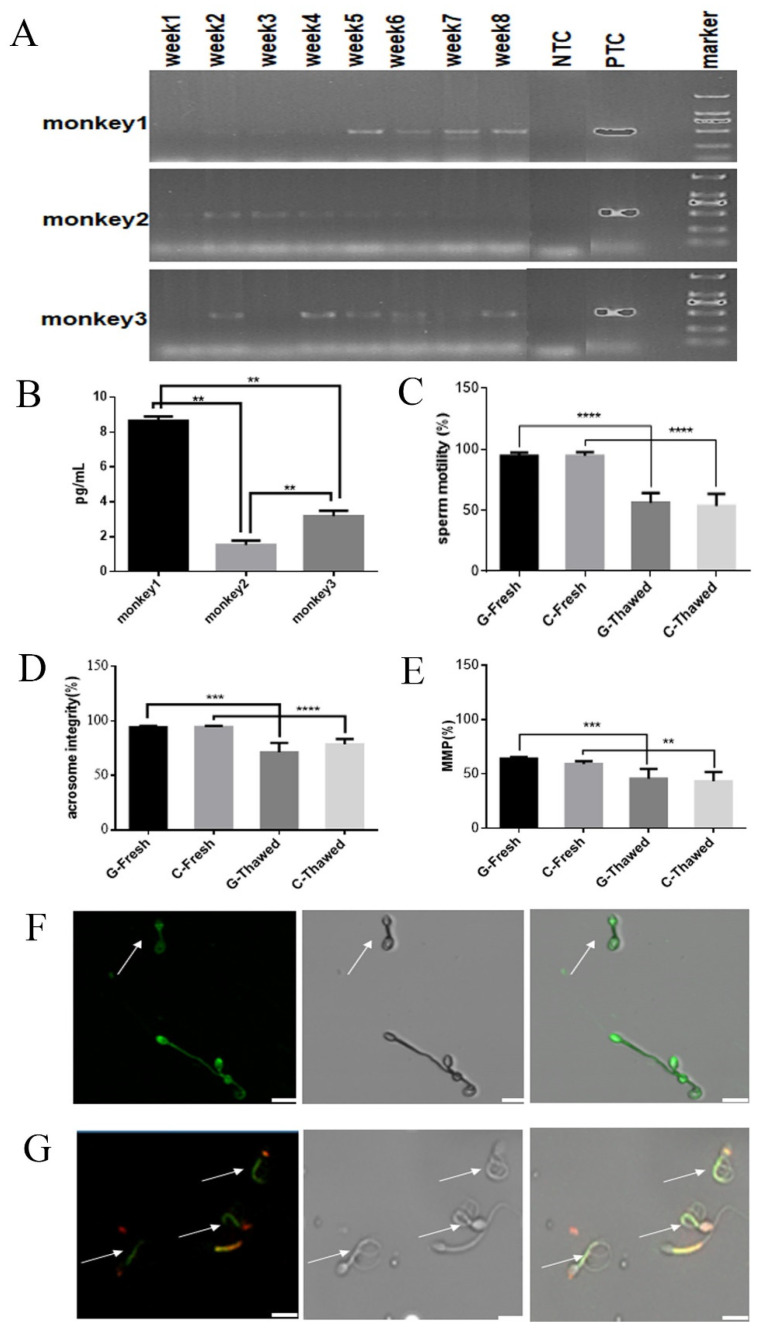
Validation of transgenic sperm and sperm evaluation before and after freezing: (**A**) PCR detection of EGFP integration: week1–week8 refer to weekly sperm collection after two months of EGFP lentivirus injection. NTC: Water replaced the sperm sample as a negative control; PTC: EGFP lentivirus replaced the sperm sample as a positive control. (**B**) q-PCR: The amount of EGFP integration in the genome; the bar chart represents the amount of EGFP expression in the 3 monkey sperms transducted with EGFP lentivirus relative to the control group. (**C**) Evaluation of sperm motility with transgenic sperm carrying EGFP gene. Eight semen samples collected from each of the three male monkeys were performed for cryopreservation and evaluation. (**D**) Evaluation of acrosome integrity. (**E**) Evaluation of mitochondrial membrane potential. (**F**) Frozen−thawed sperm cells with intact and reacted acrosomes; arrow indicates an acrosome damaged sperm. (**G**) The frozen–thawed sperm with intact and decreased mitochondria function, but no difference with normal frozen–thawed sperm; green represents sperm cells with impaired mitochondrial membrane potential, as shown by the arrow. The yellow means no impairment to the mitochondrial membrane potential of the sperm. G-Fresh: fresh sperm with transgenic sperm carrying EGFP gene; C-Fresh: control group of fresh sperm (fresh sperm with no transgenic); all data expressed as the means ± SEM. Asterisks stand for significance: ** < 0.01, *** < 0.001, **** < 0.0001, (*n* = 3).

**Figure 5 vetsci-10-00104-f005:**
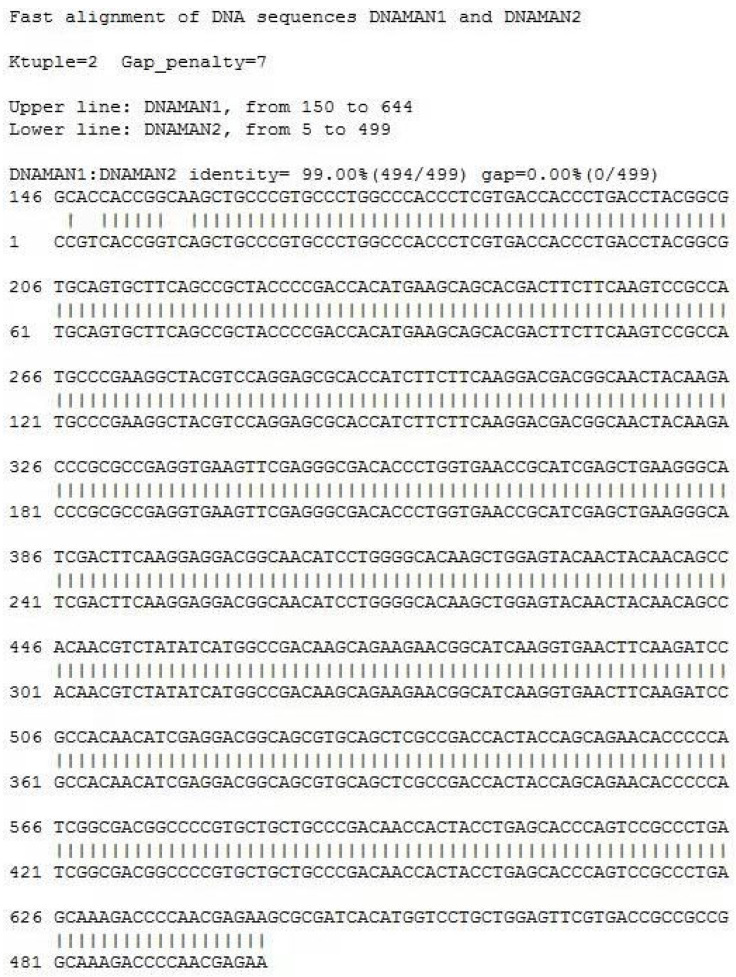
The sequence alignment of the EGFP gene was performed using first-generation sequencing techniques at the 8th week after EGFP lentivirus injection, and the alignment rate was 99%, which can prove that EGFP genes in SSCs are integrated into the sperm genome.

## Data Availability

Not applicable.

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
