# Peer review of "Generation of Transgenic Sperm Expressing GFP by Lentivirus Transduction of Spermatogonial Stem Cells In Vivo in Cynomolgus Monkeys"

_vetsci, 2023, doi:10.3390/vetsci10020104_

Round 1

Reviewer 1 Report

The authors challenged to produce transgenic sperm as a gene delivery vector by lentivirus infection to spermatogonial stem cells in cynomolgus monkey. They succussed to obtain sperm carrying the transgene with this method and found that Thy1 18 can be a surface marker of SSCs in cynomolgus monkey.

Though the present study contains some variable information, several points remain to be elucidated and revised for publication as follows.

Line 14: Please explain what SSCs indicate.

Line 69: “nonhuman primates” should be NHP.

Line 74: This would depend on the transgene and/or expression efficiency.

Line 78: “nonhuman primate” should be NHP.

Line 82: Did “Three” include experiment on trypan blue injection? None of them were sacrificed?

Line 92: “refrigerator” should be freezer.

Line 93: Is it common to use “1.0E+09” to show a number of viruses?

Line 117: Similar explanation is needed in “2.4. Pathology of testicular tissue”.

Line 118: Explanations for abbreviation “PBS” and “BSA” are needed.

Line 123: Please specify the location of Life Span Bio Sciences.

Line 125: ”Thermo” is not the exact name of the company.

Line136: Method for sperm freezing and thawing are not provided.

Line 144: Specific methods must be described.

Line 183: I feel the pictures are too many for showing no obvious change.

Line 188: Please indicate the time points for testis biopsy in Fig. 1A.

Line 203: Please provide the rate of EGFP positive sperms. Was the expression level of EGPF measured including both of EGFP positive and negative sperms, or only positive ones?

Line 216: No explanations for the color (green/red) are found.

Line 218: No explanation for asterisks is found.

Line 226-227: This sentence is duplication of the explanation in Introduction, and not necessary here.

Line 244-246: In my opinion, this is very important point, and should be stated in detail including its method. As mentioned above, the ratio of sperm with EGFP should be shown.

Line 253: The method for freezing and thawing must be described.

Line 254-255: References are needed.

Line 258-265: It is strange and inappropriate to spend half of the discussion on sperm cryopreservation, which is not the subject of this paper.

Author Response

Reviewer #1

Line 14: Please explain what SSCs indicate.

Response: We appreciated the suggestion provided from the reviewer. SSCs indicate for spermatogonal stem cells. Please see the Abstract section from Line 24 to 25.

Line 69: “nonhuman primates” should be NHP.

Response: We have revised it according to your comments, please see Line 78.

Line 74: This would depend on the transgene and/or expression efficiency.

Response: We agree that transgenic sperm and transgenic offspring depend on the transgene and/or expression efficiency. We revised the description and please see from Line 82 to 84.

Line 78: “nonhuman primate” should be NHP.

Response: We have revised it according to your comments, please see Line 88.

Line 82: Did “Three” include experiment on trypan blue injection? None of them were sacrificed?

Response: The trypan blue injection only performed on one monkey, and was not sacrificed. Please see Line 200 to 205.

Line 92: “refrigerator” should be freezer.

Response: We have revised it according to your comments, please see Line 99.

Line 93: Is it common to use “1.0E+09” to show a number of viruses?

Response: We've changed it to more common usage, 1.0×109 TU/mL. Please see Line 101.

Line 117: Similar explanation is needed in “2.4. Pathology of testicular tissue”.

Response: We have supplemented details according to your comments. Please see from Line 119 to 129.

Line 118: Explanations for abbreviation “PBS” and “BSA” are needed.

Response: We have explanations for abbreviation “PBS” and “BSA” according to your comments, please see from Line 132 to 133.

Line 123: Please specify the location of Life Span Bio Sciences.

Response: The information was added, please see from Line 138. 

Line 125: ”Thermo” is not the exact name of the company.

Response: We have revised it according to your comments, replace Thermo with Thermo Fisher Scientific, please see Line 140 to141.

Line136: Method for sperm freezing and thawing are not provided.

Response: We provide detailed sperm freezing and thawing methods according to your comments. Please see from Line 151 to 156.

Line 144: Specific methods must be described.

Response: We have provided the specific methods for acrosome integrity and mitochondrial activity evaluation, please see from Line 157 to 178.

Line 183: I feel the pictures are too many for showing no obvious change.

Response: We have revised it according to your comments, please see Line 221.

Line 188: Please indicate the time points for testis biopsy in Fig. 1A.

Response: Testicular biopsy was performed at 8 weeks after lentivirus injection, as shown in Fig. 1A "testes collection". please see Line 209.

Line 203: Please provide the rate of EGFP positive sperms. Was the expression level of EGPF measured including both of EGFP positive and negative sperms, or only positive ones?

Response: We thanks the reviewer’s comments. At present, we do not have a better solution to provide a rate of EGFP positive sperms. We can't count the rate of EGFP positive sperms under the fluorescence microscope, and the reason is the high methylation of sperm results in non-expression of the GFP protein even though the sperm carries the gene.  So here,we can only quantitatively analyze the EGFP of all sperm (both of EGFP positive and negative sperms). Please see from Line 242 to 243.

Line 216: No explanations for the color (green/red) are found.

Response: We added the indication of the colors in Figure 4 according to your comments, please see from Line 256 to 258.

Line 218: No explanation for asterisks is found.

Response: Asterisks stands for significance. We have revised it according to your comments, please see from Line 260 to 261.

Line 226-227: This sentence is duplication of the explanation in Introduction, and not necessary here.

Response: We delete this unnecessary paragraph according to your comments.

Line 244-246: In my opinion, this is very important point, and should be stated in detail including its method. As mentioned above, the ratio of sperm with EGFP should be shown.

Response: We thanks the reviewer’s comments. The fertilization method refers to previous research, please see from Line 283 to 286. At present, we can't count the rate of EGFP positive sperms under the fluorescence microscope, and the reason is the high methylation of sperm results in non-expression of the GFP protein even though the sperm carries the gene.  So here we can only quantitatively analyze the EGFP of all sperm (both of EGFP positive and negative sperms). So here we can only quantitatively analyze the EGFP of all sperm (both of EGFP positive and negative sperms). Please see from Line 242 to 243.

Line 253: The method for freezing and thawing must be described.

Response: We provide detailed sperm freezing methods according to your comments

, please see 2.6. Semen collection, evaluation and cryopreservation of Materials and methods from Line 151 to 156.

Line 254-255: References are needed.

Response: We have added references based on your comments, please see Line 291.

Line 258-265: It is strange and inappropriate to spend half of the discussion on sperm cryopreservation, which is not the subject of this paper.

Response: We appreciate your advice and we have shortened the unrelated description and revised the discussion. Please see Line 288 to 302.

Reviewer 2 Report

The authors described a method in order to obtain transgenic sperm. The Introduction and the MM are clearly presented as well as the results. In the Discussion seciton the authors may wish to elaborate about the method and whether this method can be a useful tool for humans

Author Response

Reviewer #2

The authors described a method in order to obtain transgenic sperm. The Introduction and the MM are clearly presented as well as the results. In the Discussion seciton the authors may wish to elaborate about the method and whether this method can be a useful tool for humans.

Response: Thank you very much for your comments.  This approach is expected to become another effective way to establish gene edited animal disease models, and play an important role in the research of male reproductive diseases and the development of new drugs and therapeutic methods. Please see Line 307 to 310.

Reviewer 3 Report

Dear Authors,

In my opinion, the manuscript submitted for review is interesting and innovative. The authors presented the possibility of transgenesis in NHPs and obtained transgenic spermatozoa, which were cryopreserved and evaluated for their biological properties. This last part of the work needs significant improvement.

My comments mainly concern the description of methods related to the assessment of sperm quality parameters: motility, acrosome integrity and mitochondrial activity:

It should be specified in the description which method of assessing motility - subjectively or using the CASA system?

Information on the assessment of acrosome integrity and mitochondrial activity should be completed. Were these parameters evaluated using a microscope or a flow cytometer? How were the samples prepared?

In addition, the number of samples used for the tests, the results of which are presented in the graphs (n=?), should be provided.

Statistical analysis should be included in the description of the methods. Please provide the statistical program and statistical tests that were used in this analysis. It is insufficient to provide information that the statistical analysis has been described in previous article.

Minor Notes:

Line 5. Remove the preposition and ’ at the end of the listed authors, and put it before the name of the last author.

Figure 5 should be corrected, as the presented figures, especially B,C,D and E, are poorly visible. I suggest posting photos separately - additional Figure.

Author Response

Reviewer #3

In my opinion, the manuscript submitted for review is interesting and innovative. The authors presented the possibility of transgenesis in NHPs and obtained transgenic spermatozoa, which were cryopreserved and evaluated for their biological properties. This last part of the work needs significant improvement.

My comments mainly concern the description of methods related to the assessment of sperm quality parameters: motility, acrosome integrity and mitochondrial activity:

It should be specified in the description which method of assessing motility - subjectively or using the CASA system?

Response: Currently, the CASA systems are designed for human, livestocks and rodents. There is no CASA chips and software specifically for non-human primates. We put fresh sperm and thawed sperm samples (10μL) were on a pre-warmed Makler counting chamber (Sefi Medical Instruments, Haifa, Israel) under a microscope for motility assessment. At least 200 sperm of each sample were evaluated for motility and repeated eight times. This method for motility evaluation of monkeys have been widely reported,and the samples are listed below for reference.

  1. Feradis, A.H.; Pawitri, D.; Suatha, I.K.; Amin, M.R.; Yusuf, T.L.; Sajuthi, D.; Budiarsa, I.N.; Hayes, E.S. Cryopreservation of epididymal spermatozoa collected by needle biopsy from cynomolgus monkeys (Macaca fascicularis). Journal of medical primatology 2001, 30, 100-106, doi:10.1034/j.1600-0684.2001.300205.x.
  2. Chen, B.; Li, S.; Yan, Y.; Duan, Y.; Chang, S.; Wang, H.; Ji, W.; Wu, X.; Si, W. Cryopreservation of cynomolgus macaque (Macaca fascicularis) sperm with glycerol and ethylene glycol, and its effect on sperm-specific ion channels - CatSper and Hv1. Theriogenology 2017, 104, 37-42, doi:10.1016/j.theriogenology.2017.08.009.

Information on the assessment of acrosome integrity and mitochondrial activity should be completed. Were these parameters evaluated using a microscope or a flow cytometer? How were the samples prepared?

Response: We thanks the reviewer’s comments. We have provided information on the assessment of acrosome integrity and mitochondrial activity of sperm. We used fluorescence microscope for the evaluation, please see Line 157 to 178.

In addition, the number of samples used for the tests, the results of which are presented in the graphs (n=?), should be provided.

Response: We have added the information, please see Line 215, 225, 234, 261.

Statistical analysis should be included in the description of the methods. Please provide the statistical program and statistical tests that were used in this analysis. It is insufficient to provide information that the statistical analysis has been described in previous article.

Response: We provide the description about the Statistical analysis according to your comments, please see 2.8. Semen collection, from Line 189 to 194.

Minor Notes:

Line 5. Remove the preposition ‘and ’ at the end of the listed authors, and put it before the name of the last author.

Response: We thanks the reviewer’s comments. We have revised it according to your comments, please see Line 5.

Figure 4 should be corrected, as the presented figures, especially B, C, D and E, are poorly visible. I suggest posting photos separately - additional Figure.

Response: We thanks the reviewer’s comments. We have improved the quality of these figures, please see supplemental FigureS1-S4, from Line 440 to 468.     

Round 2

Reviewer 1 Report

The authors carefully revised the most parts of the manuscript according to the reviewer’s comments. However, there are still some points to be revised as follows:

Line 82: Did “Three” include experiment on trypan blue injection? None of them were sacrificed?

Response: The trypan blue injection only performed on one monkey, and was not sacrificed. Please see Line 200 to 205.

> Does “one monkey (line 200)” mean one of three monkeys (n=3, line 91) or extra one addition to the three monkeys? If the former, the monkey should be injected with EGFP lentivirus in one testis after surgical excision of the other testis injected trypan blue. Didn’t this affect the results of subsequent sperm analysis? If the latter, total number of the monkeys used in the present study, that is 4, should be clearly stated for ethical reasons.

Line 188: Please indicate the time points for testis biopsy in Fig. 1A.

Response: Testicular biopsy was performed at 8 weeks after lentivirus injection, as shown in Fig. 1A "testes collection". please see Line 209.

> It is confusing because there is no statement of “testes collection” in the text, and this term recalls surgical excision of the testis like trypan blue experiment. Why not unify the terms with “biopsy”?

Line 283-86: As mentioned before, this is very important point. Why the authors did not inject EGFP lentivirus into testes of  both sides to raise the ratio of sperm carrying EGFP gene. Please discuss this poit.

Line 258-265: It is strange and inappropriate to spend half of the discussion on sperm cryopreservation, which is not the subject of this paper.

Response: We appreciate your advice and we have shortened the unrelated description and revised the discussion. Please see Line 288 to 302.

> If the authors wish to discuss about sperm cryopreservation, they should explain its importance in Introduction section.

Minor points

Line 138, 140, 163 etc.: The writing ways of companies' locations are not unified.

Line 154: “smoke” and vapor are different.

Author Response

Reviewer #1

Line 82: Did “Three” include experiment on trypan blue injection? None of them were sacrificed?

Response: The trypan blue injection only performed on one monkey, and was not sacrificed. Please see Line 200 to 205.

> Does “one monkey (line 200)” mean one of three monkeys (n=3, line 91) or extra one addition to the three monkeys? If the former, the monkey should be injected with EGFP lentivirus in one testis after surgical excision of the other testis injected trypan blue. Didn’t this affect the results of subsequent sperm analysis? If the latter, total number of the monkeys used in the present study, that is 4, should be clearly stated for ethical reasons.

Response:  Total number of the monkeys used in the present study is 4. We have supplemented details according to your comments.  Please see Line 97 to 100.

Line 188: Please indicate the time points for testis biopsy in Fig. 1A.

Response: Testicular biopsy was performed at 8 weeks after lentivirus injection, as shown in Fig. 1A "testes collection". please see Line 209.

> It is confusing because there is no statement of “testes collection” in the text, and this term recalls surgical excision of the testis like trypan blue experiment. Why not unify the terms with “biopsy”?

 Response: We thanks the reviewer’s comments. It is testicular biopsy. We have supplemented details according to your comments.  Please see Fig 1A.

Line 283-86: As mentioned before, this is very important point. Why the authors did not inject EGFP lentivirus into testes of both sides to raise the ratio of sperm carrying EGFP gene. Please discuss this poit.

 Response: Sorry for the confusion. We injected both sides. Please see the revision of line115.

Line 258-265: It is strange and inappropriate to spend half of the discussion on sperm cryopreservation, which is not the subject of this paper.

Response: We appreciate your advice and we have shortened the unrelated description and revised the discussion. Please see Line 288 to 302.

> If the authors wish to discuss about sperm cryopreservation, they should explain its importance in Introduction section.

Response: We appreciate your advice and the importance about sperm cryopreservation was added in Introduction section. Please see Line 79 to 83.

Minor points

Line 138, 140, 163 etc.: The writing ways of companies' locations are not unified.

Response: We have unified the writing ways of companies' locations, please see Line 145, 147, 170, 178.

Line 154: “smoke” and vapor are different.

Response: We have revised it according to your comments, replace smoke with vapor, please see Line 161 to162.

Reviewer 3 Report

Dear Authors,

The manuscript has been corrected in accordance with the comments.

I have one major note:

Was the statistical analysis for sperm quality parameters based on only three samples? Were there more repetitions?

I have some minor comments:

The sentence "Briefly, the semen was evenly coated with 10μl fresh semen and frozen thawed semen on a slide..................." is not entirely clear. Please correct the translation (lines 163-164).

JC-1 staining assesses the mitochondrial membrane potential, not "the integrity of the mitochondrial membrane potential" as stated in the manuscript (line 172).

Minor errors in the text of the manuscript should also be noted, e.g.

instead of "105" it should be "105" (line 173).

Author Response

Reviewer #3

I have one major note:

Was the statistical analysis for sperm quality parameters based on only three samples? Were there more repetitions?
Response: Eight semen samples collected from each of the three male monkeys were performed for cryopreservation and evaluation. Please see Line 153 to 154.

I have some minor comments:

The sentence "Briefly, the semen was evenly coated with 10μl fresh semen and frozen thawed semen on a slide..................." is not entirely clear. Please correct the translation (lines 163-164).

Response: We have corrected the translation. Please see Line 170 to 172.

JC-1 staining assesses the mitochondrial membrane potential, not "the integrity of the mitochondrial membrane potential" as stated in the manuscript (line 172).

Response: We have corrected the translation. Please see Line 179.

Minor errors in the text of the manuscript should also be noted, e.g.

instead of "105" it should be "105" (line 173).

Response: We have corrected the information. Please see Line 179.